# Electrically responsive photonic crystals with bistable states for low-power electrophoretic color displays

Qianqian Fu[1,2], Wenyuan Yu[1,2], Guangyang Bao[1,2] & Jianping Ge [1,2] ✉

Electrically responsive photonic crystals are promising materials for electrophoretic color displays with better brightness and color saturation. However, electric field must always be applied to maintain the specific colors, which brings concerns about the power consumption and signal stability and reversibility. Here, we show an electrically responsive photonic crystal with two stable states at 0 V, which are the colored state or the colorless state with ordered or disordered particle arrangement. The color state can be reversibly switched by applying a short-time electrical field, just like in the case of commercial electrophoretic ink. With optimized recipe and electric field, the photonic crystals encapsulated in the prototype display panel are proved to have potentials in high resolution, multi-color, and greyscale display, which lays down a firm basis for reflective displays with low power consumption and good visibility.

Electrophoretic displays (EPDs) are devices that mimic the appearance of ordinary ink on paper. They have attracted great attention due to their broad applications in E-book readers, wristwatches, mobile phones, electronic tags, and timetables at the bus station[1–5]. Unlike conventional displays that emit light, EPD reflects ambient light like paper, which makes them more comfortable to watch and gives more protection to our eyes. More importantly, EPD can hold static text or images without electricity, which consumes much less power than emissive displays. The commercialized electrophoretic inks are usually composed of highly scattering and absorbing microparticles with opposite charges in an electrophoretic medium. When a positive or negative electric field is applied, one of these two microparticles will migrate to the front side of the display, thus giving the pixel a white or black appearance. As the ink continues to evolve, there appears to be increasing interest in the color EPDs in these years[6–8]. One strategy for color EPD is the preparation of special inks composed of three to four colorant particles with different surface charges. But the precise control of the particle charges in synthesis and the control of the electric field in use will be great challenges[7,9,10]. Another strategy is the coating of color filters in front of the electrophoretic cells, but the brightness and saturation of the as-produced colors are unsatisfactory[11].

The electrically responsive photonic crystal (ERPC) based on nanoparticle electrophoresis could be a promising material for next-generation color EPD due to its tunable structural colors and compatibility with the current technology[12]. When the electric field changes, the charged nanoparticles in ERPCs will migrate towards or away from the front transparent electrode, leading to the shrinkage or expansion of the PC lattice and thereby the change of colors. Saturated colors across the visible range can be achieved from a single colloidal material by tuning the E-field strength. Unlike the other ERPCs based on electrochemical reactions[13–16], liquid crystal reorientation[17], electrostriction of PC elastomers[18–21], and electrothermal PC hydrogels[22,23], the electrophoretic ERPCs possess similar material compositions and working conditions as the conventional electrophoretic ink, which makes it fully compatible with the encapsulation crafts, the control circuit, and other mature technologies.

Since the first report of electrophoretic ERPC in 2010[24], substantial research has been performed for the investigation of new colloidal materials[24–28] and dispersion medium[29–32], the enhancement of particle surface charge[33,34], and the modification of electrodes[35,36], all of which are adopted to improve the color tuning range, the responding speed, the stability, and the reversibility of ERPC. However,

[1]School of Chemistry and Molecular Engineering, Shanghai Key Laboratory of Green Chemistry and Chemical Processes, East China Normal University, 200062 Shanghai, China. [2]Institute of Eco-Chongming, 202162 Shanghai, China. ✉e-mail: jpge@chem.ecnu.edu.cn

there still has a critical difficulty, that is, the bistability, to conquer before the ERPC truly becomes a usable PC material for EPD. Up till now, all the electrophoretic ERPC only has one stable state, in which the particles spontaneously assemble into colloidal crystals in the absence of an E-field. Once the ERPC is compressed to show other colors under specific electric fields, the field must always be applied to generate electrophoretic packing force to balance the increased net repulsion from the neighboring particles so that the compressed crystal lattices and those structural colors can be maintained. The continuous exertion of electric fields not only increases power consumption but also leads to electrochemical reactions and unstable optical signals, which significantly restrict its application in the electrophoretic display.

In this work, we report an ERPC with bistable and switchable characteristics and develop an electrophoretic ink for color display. Unlike the conventional ERPC, the $SiO_2$/PCb-PEG-EG ERPC is unable to show abundant colors at different voltages due to its weak color tunability. However, in addition to the stable colored state from colloidal self-assembly, it also has a stable colorless state at 0 V with disordered particle stacking. Because the introduction of polyethylene glycol (PEG) increases the viscosity to balance the electrostatic repulsion, which further freezes the particles' Brownian motion and stabilizes the disordered structure. More importantly, these two states can also be reversibly switched by application of a short-time electrical field, just like in the case of commercial electrophoretic ink. With all these features, an ERPC based on colloidal assembling and disassembling is proposed, where the bistability of optical signals and reversibility in electrical response makes it an ideal material for EPD. In the following discussion, the recipe of bistable ERPC and the waveform of the electrical field are studied to realize the long-time holding of the colorless state and the quick recovery of the colored state. Furthermore, the potential of bistable ERPC for high-resolution, multi-color, and greyscale displays is also investigated by prototype display panels. It is believed that the bistable ERPC can be used as next-generation color electrophoretic ink and certainly will find more application in reflective displays with low power consumption and good visibility.

## Results

### ERPC with bistable and switchable characteristics

The ERPC with bistability was prepared by the supersaturation-induced precipitation and crystallization of $SiO_2$ particles in the mixture of propylene carbonate (PCb), polyethylene glycol (PEG), and ethylene glycol (EG)[37]. Here, the $SiO_2$ particles with negative surface charge and narrow size distribution were used as the building blocks to construct the liquid PCs and output structural colors. PCb was chosen as a polar organic dispersion medium because it provided a dielectric environment for colloidal assembly and electrophoresis and avoided the electrochemical side reactions associated with the aqueous colloidal system. Different from all the previous ERPCs, PEG was introduced to increase the viscosity of the dispersion and balance the interparticle repulsion, which helped to freeze the particles' Brownian motion and stabilize the disordered packing structure. Additionally, EG was added to enhance the electrostatic repulsion between particles and promote colloidal reassembly. For ease of comparison, the composition of all ERPCs in the current work was summarized in Supplementary Table 1.

The as-prepared $SiO_2$/PCb-PEG-EG ERPC possessed bistable states in the absence of an electric field, which was the colored state with the highly ordered particle arrangement and the colorless state with disordered particle stackings. (Fig. 1a–e) Driven by the increase of total entropy of the colloidal system, the $SiO_2$ particles in the homogeneous solution would spontaneously precipitate to form liquid PCs with structural colors, such as green here, which led to a colored state for the system. A corresponding scanning electron microscope (SEM) image of the ERPC after freeze-drying confirmed the ordered particle arrangement, and the strong reflection or transmission peak around 570 nm was also consistent with the green color and microscopic structures. When an electric field was applied, the $SiO_2$ particles would move towards and enrich around the anode to form disordered particle stacking and maintain it even after withdrawing the voltage, which led to a colorless state for the system. SEM image confirmed the disordered structure, and the flat reflection and transmission curves around 0% and 90% were consistent with the colorless and transparent appearance. It should be noted that the sample showed a black

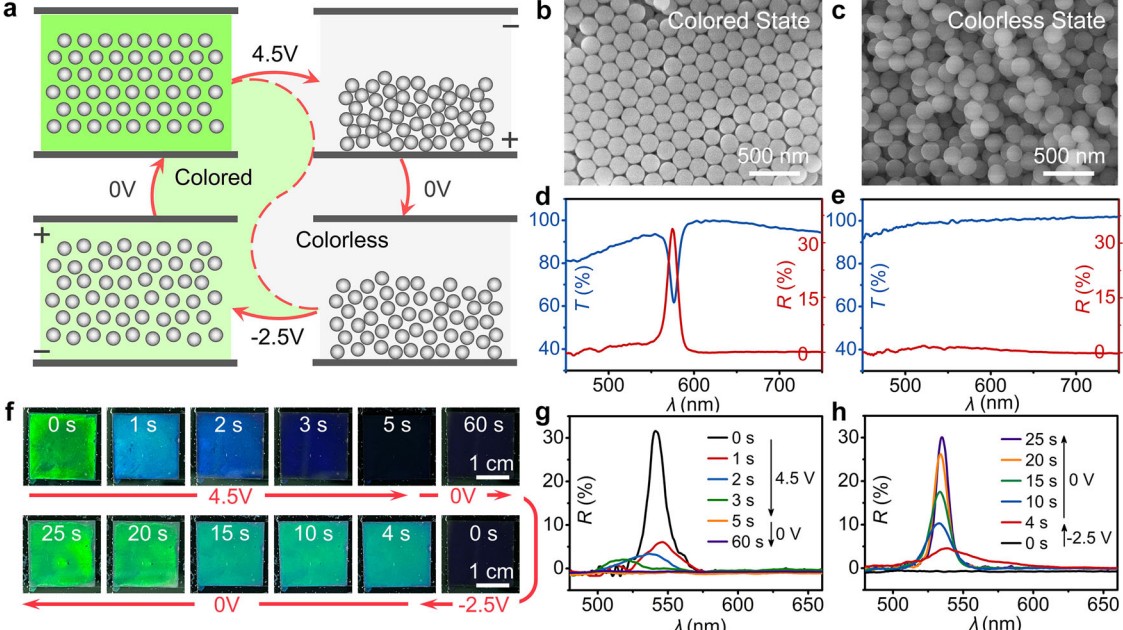

**Fig. 1 | ERPC with bistable and switchable characteristics. a** Structure change of the $SiO_2$/PCb-PEG-EG ERPC (30/60/5/5 vol%) as it was switched between the colored state and the colorless state by an electric field; **b**, **c** SEM images, and **d**, **e** the reflection (red plots) and the transmission spectra (blue plots) for the ERPC in the colored state and the colorless state; **f** time evolution of the structural colors and **g**, **h** the reflection spectra during a complete circle of color switching.

appearance in the colorless state because a light absorber, such as a 3 M insulation tape, was attached beneath the PC to reduce the incoherent light scattering. In the colorless state, both the high particle volume fraction ($f_{SiO2}$) and the PEG component contributed to the increase of viscosity and the retarding force, which froze the Brownian motion of particles and stabilized the disordered structure.

The aforementioned colored and colorless states could be reversibly switched by the exertion of a short-time electric field. (Fig. 1a, 1f–h) When a voltage of 4.5 V was applied to the SiO₂/PCb-PEG-EG ERPC for 5 s, the negatively charged SiO₂ particles moved to the bottom electrode immediately and formed a highly compressed disordered stacking structure. Since the $f_{SiO2}$ quickly increased to 50% in this case, the viscosity of the colloidal system was significantly increased to 11.9 folds of the viscosity of the mixing solvent ($\eta_{solvent}$). (Supplementary Note 1, 2) Although the particle stacking had a slight expansion after the electric field was removed, it still kept the disordered structure due to the high viscosity and freezing of Brownian motion, so the ERPC eventually turned from the colored to the colorless state. The photos and OM images (Supplementary Fig. 1) showed that the color changed from green to black in 5 s and kept black until 60 s. Correspondingly, the reflection peak blue-shifted and disappeared in 5 s and kept disappearance until 60 s.

When an inverse voltage of −2.5 V was applied to the ERPC for 4 s, the particles migrated to the top electrode, and the particle stacking expanded to a quasi-ordered structure rapidly, which gradually recovered to the highly ordered arrangement after the electric field was removed. It should be noted that the exertion of inverse voltage was necessary because the $f_{SiO2}$ quickly decreased to 30% for a quasi-ordered particle arrangement and the viscosity of the colloidal system decreased to 2.7 folds of $\eta_{solvent}$ accordingly. (Supplementary Note 1, 2) As a result, the viscous retarding force was no longer strong enough to lock the particles' motions, which let them assemble back into an ordered structure. During the switching from the colorless to colored state, the ERPC usually recovered the structural color as soon as the voltage of −2.5 V was applied but required tens of seconds under 0 V to generate intense green color, which could be proved by the photos and OM images. In the reflection spectra, a weak peak around 538 nm appeared in 4 s and evolved to a strong one in 25 s, which also confirmed the structure and color changes.

According to the bistable and switchable characteristics, a working mechanism based on "assembling and disassembling" could be proposed to define a different kind of electrophoretic ERPC. (Supplementary Fig. 2) For the traditional ERPC, the structural color was adjusted by the "shrinkage and expansion" of the PC lattice under electric fields with different voltages. While, for the bistable ERPC, the color was turned on/off through the assembling/disassembling of colloidal PCs triggered by the electric field. Although the assembling/disassembling was controlled by the viscosity and the viscosity usually decreased with the environmental temperature, the current ERPC still possessed bistability and switching ability when the temperature was kept between 25 °C and 60 °C, which ensured the practical application of bistable ERPC. (Supplementary Fig. 3).

The working mechanisms suggested that both traditional and bistable ERPCs worked through the electrophoresis of colloidal particles but not an electrochemical process (Supplementary Fig. 4) under an electric field. The major difference was that the traditional ERPC required a continuous electric field to keep its color, and the bistable ERPC did not. For the traditional SiO₂/PCb ERPC, the reflection wavelength changed from 674 nm to 540 nm as the voltage increased from 0 V to 3 V, and 3 V of voltage must always be applied to stabilize the reflection signal. In contrast, the bistable SiO₂/PCb-PEG-EG ERPC possessed stable reflection at 548 nm or no reflection at all under 0 V, where a 4.5 V or −2.5 V of voltage was applied to trigger the switching. Such bistable and switchable characteristics completely solved the problems of the unstable optical signal and high-power consumption

for traditional ERPC, which made the current ERPC ideal material for the construction of an electrophoretic display.

## Optimizing ERPC's recipe and E-field for better performance

According to the assembling-disassembling mechanism, the key to the bistability was the long-time holding of the disordered particle stacking and the quick recovery of the ordered colloidal crystals. In the colorless state with the disordered structure, the highly charged particles were usually packed very tightly near the electrode. The strong electrostatic repulsion between the particles tended to separate them from each other and assemble them back into the colloidal crystals so that it was difficult to hold the disordered structure. This was also the reason for the requirement of continuous E-field for traditional ERPC. When the introduction of viscous medium addressed the above concern, it also brought another problem, which was the slowing of colloidal assembly and recovery of the colored state. As a result, the recipe of the ERPC and the waveform of the E-field had to be optimized to balance the opposite influences and achieve a good bistability performance.

First of all, the introduction of polyethylene glycol could effectively increase the viscosity of the colloidal system and help to hold the colorless state for a long period. (Fig. 2a–c) Here, 4.5 V of voltage was applied to the SiO₂/PCb-PEG ERPCs with different molecular weights and volume fractions of PEG. Time evolution of reflection intensities (R-t curve) was plotted, where a low recovery rate for reflection intensity ($\Delta R/t$) indicated a better holding of the colorless state. (Supplementary Fig. 5, 6) When the fraction of PEG-1500 was increased from 2.5% to 10%, $\Delta R/t$ reduced from 0.070 s⁻¹ to 0.011 s⁻¹, and it further decreased to 0.003 s⁻¹ if 10% of PEG-20000 was used, which suggested that both the high fraction and the large molecular weight were favorable to the stabilization of disordered state. (Supplementary Table 2) This is because the kinematic viscosity of the PCb-PEG mixtures (Supplementary Table 3) increased with the increase of PEG's volume fraction and molecular weight so that the viscous force became large enough to balance the strong electrostatic repulsion between particles, which froze the Brownian motion and maintained the disorderly stacking structure. However, an over-viscous medium made it difficult for the particles to reassemble (Supplementary Fig. 7), and 5% of PEG-20000 was the optimized recipe for the bistable ERPC.

Since the high viscosity slowed down the recovery of the colored state, ethylene glycol was introduced to enhance the interparticle repulsion and accelerate the colloidal reassembling. (Fig. 2d–f, Supplementary Fig. 8) The introduction of EG has two functions here. Firstly, EG contributed to the holding of the colorless state because the addition of EG significantly increased the viscosity of the ERPC in the presence of PEG and SiO₂ particles. (Supplementary Fig. 9) However, EG could not replace all the PEG in ERPC, as the SiO₂/PCb-EG colloidal system was inadequately viscous to maintain the disordered state. (Supplementary Fig. 10) More importantly, EG accelerated the recovery of the colored state after an inverse voltage of −2.5 V was applied. When 5% of EG replaced the same amount of PCb in the ERPC, $\Delta R/t$ increased from 0.604 s⁻¹ to 1.029 s⁻¹, and the switching time was shortened to 35 s. (Supplementary Table 4) Such improvement might be related to the solubility property of EG. As learned from the Hansen solubility parameters, EG had a much higher hydrogen bond parameter ($\delta_h$) than PCb, which meant a stronger combination with Si-OH groups on the SiO₂ particles. The enhanced solvation led to the enhancement of interparticle repulsion and then accelerated the reassembling process. Similar to the introduction of PEG, an excessive fraction of EG, such as 10%, was unfavorable to the recovery of the colored state due to high viscosity, so 5% of EG was probably the optimized condition.

Furthermore, an effective switching between the colored and colorless state could be achieved only at a proper particle volume fraction ($f_{SiO2}$). (Fig. 2g–i) After the exertion of 4.5 V to the ERPC, a low

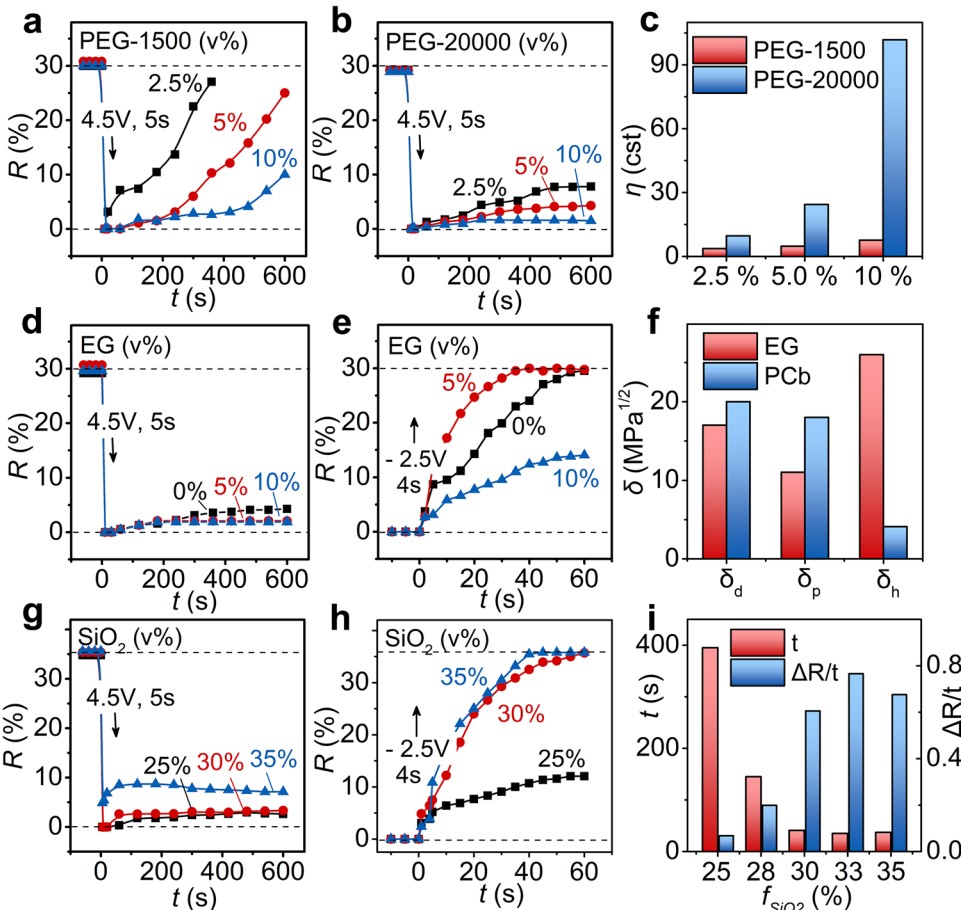

**Fig. 2 | Optimizing the recipe of bistable ERPC. a, b** Influence of PEG volume fraction and molecular weight on the holding of colorless state, indicated by the time evolution of reflection intensities and explained by **c** the measured viscosities of PCb-PEG mixtures; **d, e** influence of EG volume fraction on the holding of colorless state and quick recovery of colored state, explained by **f** the solubility parameters; **g, h** influence of volume fraction of $SiO_2$ particles on bistability, and **i** assembly time and efficiency ($\Delta R/t$) of corresponding ERPCs without electric field. The recipes of the above ERPCs are listed in Supplementary Table 1.

$f_{SiO2}$ seemed to be favorable to the stabilization of the disordered colorless state because the ERPC had a strong tendency to recover part of the crystalline structure once the $f_{SiO2}$ was increased above 30%. (Supplementary Fig. 11, 12) On the other hand, when the reassembly of colloidal crystals was triggered by a voltage of −2.5 V, a high $f_{SiO2}$ was then beneficial to the quick recovery of the colored state because the particles tended to precipitate out at high concentration to form colloidal crystals. This could also be confirmed by the assembly time for the liquid PCs in the absence of any electric field, which decreased from 395 s to 37 s as the $f_{SiO2}$ increased from 25% to 35%. (Supplementary Fig. 13) In summary, for a balance between the holding of bistable states and their switching, the recipe of the bistable ERPC was optimized to 30% of $SiO_2$ particles, 60% of PCb, 5% of PEG-20000, and 5% of EG.

In addition to the recipe of ERPC, the electric field was also modulated in the form of specific waveforms to enhance the holding of disordered state and accelerate the recovery of ordered state at the same time. It should be noted that the following E-field modulations usually had a similar influence on ERPCs with different recipes. Because such modulation functioned through the time-resolved electrophoretic forces, which were seldom affected by the particle size or small change of solvent's volume fractions. In the switching from the colored state to the colorless state, a long-time stabilization of the disordered structure could be achieved by increasing the E-field strength from 4.0 V to 5.0 V (Fig. 3a, d) or prolonging the E-field duration time from 5 s to 20 s at 4.5 V (Supplementary Fig. 14). Under strong electric field conditions, the assembly of particles could be completely disrupted, resulting in structures that were more disordered and more difficult to restore ordered arrangement. However, too high voltages such as 5.0 V or too long exertion such as 20 s might cause electrochemical reactions in the system, which made it difficult to reassemble the particles. Therefore, an intermittent E-field modulation was adopted, where 4.5 V of voltage was applied for 5 s in every 60 s. (Fig. 3b, e) Compared to the result of a one-time application of 4.5 V for 20 s, the bistable ERPC under intermittent modulation could maintain the colorless state very well in 300 s yet recover to the colored state after applying the inverse voltage. (Supplementary Fig. 15) The assembling and disassembling mechanism of the current ERPC determined that the long-term holding of disordered state and quick recovery of ordered state were contradictory requirements. The intermittent modulation actually offered a balanced solution by constructing an "active" disordered state so that the colorless ERPC always got itself ready to recover to the colored state under inverse voltage.

In the switching from the colorless state to the colored state, an inverse E-field with a more precise control was required to speed up the recovery of the ordered structure. (Fig. 3c, f) Generally, a stronger inverse voltage or a longer E-field application led to quicker recovery of the colored state because both of them facilitated the redispersion of densely packed $SiO_2$ particles and accelerated the restoration of homogeneous colloidal dispersion with $f_{SiO2}$ of 30%. For example, when the inverse voltage was enhanced from −1.5 V to −3.5 V, or when the exertion of −2.5 V was extended from 2 s to 4 s, the recovery time indeed decreased accordingly. (Supplementary Fig. 16 and 17) However, an excessively strong or long inverse voltage brought negative

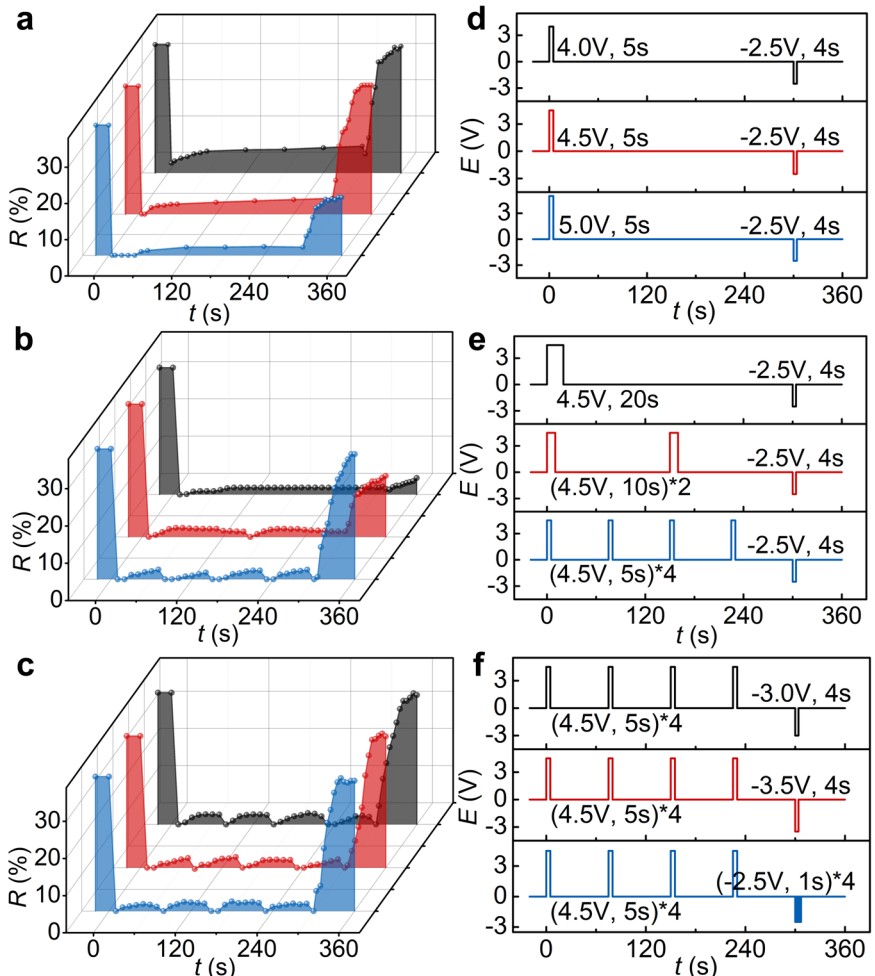

**Fig. 3 | Modulation of the applied E-field.** The bistability of SiO$_2$/PCb-PEG-EG ERPCs (30/60/5/5 vol%) evaluated by **a**–**c** the time evolution of reflection intensity under different E-field waveforms, including **d** a strength modulation and **e** an intermittent modulation of the forward voltage, and **f** an impulsive modulation of the inverse voltage. The black, red, and blue reflection intensity evolutions in **a**–**c** were recorded under the corresponding E-fields in **d**–**f** plotted by the same color.

effects instead, as it would lead to the compressing of the just recovered colloidal crystal towards the opposite electrode. Such inverse compression caused by 4 s of −4.5 V or 6 s of −2.5 V led to a decrease of reflection intensity and wavelength compared to original values. Therefore, 4 s of −2.5 V was a proper inverse voltage that could be used to recover the colored state in 45 s. Moreover, an impulsive modulation of the inverse voltage at −2.5 V with application of 1 s and interval of 0.5 s could further shorten the recovery time to 31 s, probably because the exerted coulombic force was more favorable to the redispersing of particles and recovery of colloidal crystals.

With optimized recipe and E-field modulation, the bistable display and reversible switching of the ERPC could be realized under general working conditions. (Supplementary Fig. 18, 19) For the application in the outdoor display, such as bus board, the electrophoretic ink usually needs to be refreshed within one minute. In a simulated test, the bistable ERPC could exhibit colored and colorless states alternatively and switch to each other every 60 s, which could be confirmed by the change of reflection intensity with time. Although the switching from the colorless to the colored state took about 30 s to recover the full reflection intensity, the perceptible switching might be faster as the structural color appeared just several seconds after the trigger by E-field, and the reflection wavelength showed little changes in the following time. (Supplementary Fig. 20) In an extended test with 30 cycles of switching, the reflection wavelength of the colored state was almost unchanged while its intensity changed periodically, which

further confirmed the effective refreshing and good reversibility of the ERPC. For the application in E-tags, both bistability and switching performance in the long-term are essential. The current ERPC could maintain the colored or colorless state for 30 min yet still be able to switch to each other quickly. These cycling tests covered most application scenarios of the bistable ERPC, which demonstrated its potential for the reflective display.

## Reflective display units based on bistable ERPCs

Electrophoretic display units with micron-scale accuracy could be achieved through the fabrication of cell arrays or patterned ITO electrodes by laser micromachining. Here, a series of strip-like display units with widths ranging from 250 μm to 500 μm were fabricated by encapsulating the green ERPC in the epoxy frames prepared by the laser engraver. (Supplementary Fig. 21) In the colored state, all the ERPCs showed bright structural colors, and they were all composed of a large number of colloidal microcrystals, which suggested the assembly of SiO$_2$ colloidal PCs was not affected by the cell size in this range. The measured unit widths were close to the designed values, and the practical resolution considering the width of both cells and intervals was determined to vary from 25 to 51 lines per inch (LPI). Furthermore, a linear scanning across the display units provided spatially resolved microscopic reflection spectra, which also confirmed the LPI resolutions that could be achieved. Here, the reflection intensity decreased at high resolutions because the strip-like unit became

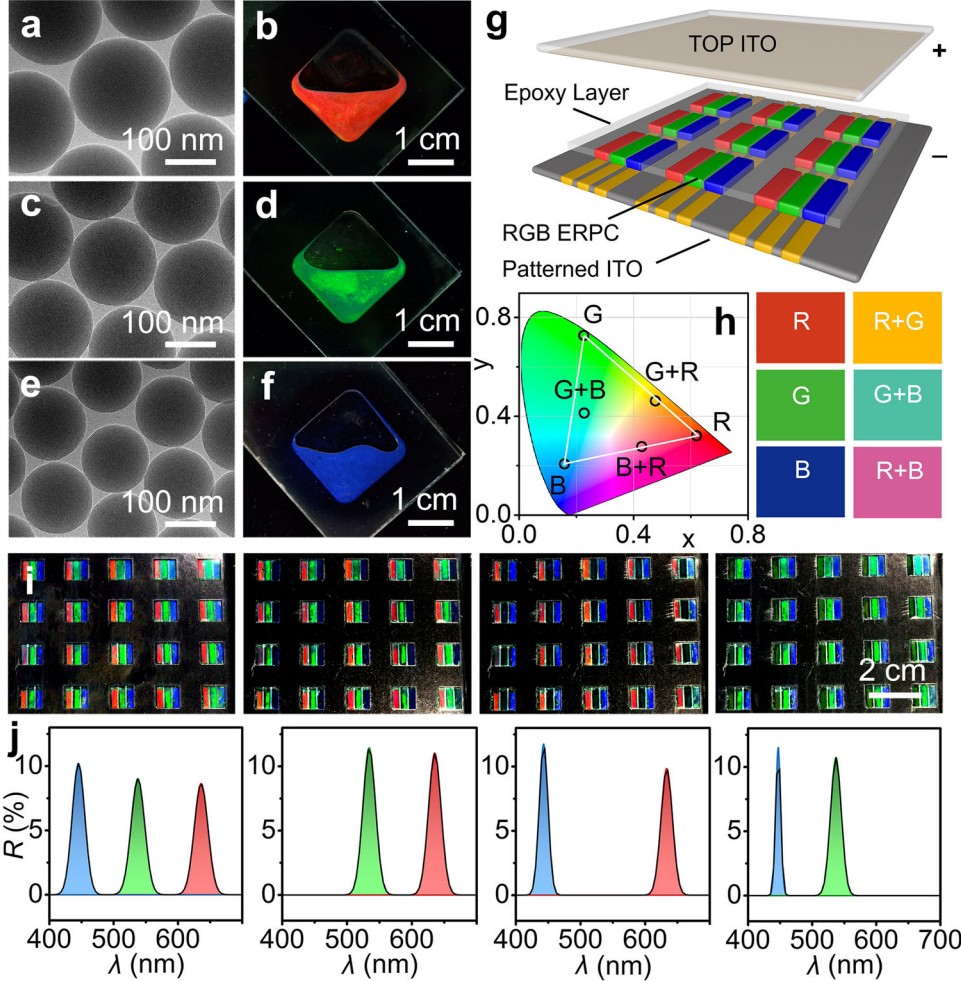

**Fig. 4 | Multi-color display based on bistable ERPCs. a–f** TEM images and photos of red, green, and blue SiO$_2$/PCb-PEG-EG ERPCs (30/60/5/5 vol%) prepared by SiO$_2$ particles with diameters of 199 nm, 169 nm, and 138 nm; **g** schematic diagram of a prototype display panel composed of RGB pixel arrays; **h** the chromaticity coordinates of the primary colors and the mixing colors; **i** digital photos of RGB pixel arrays showing different colors; and **j** the corresponding reflection spectra of a single pixel.

smaller than the sampling range of the spectrometer probe. In addition to the encapsulation of ERPC in microgrooves, the high-resolution display could also be achieved by a patterned ITO electrode, where the ERPCs were no longer required to be loaded in separate cells. (Supplementary Fig. 22) It should be noted that we tended to discuss the display accuracy and resolution limit of the ERPC material, but it was inevitably affected by the manufacturing crafts, such as the resolution of the laser engraver here. Since the colloidal microcrystals usually had a size from several to tens of micrometers, it was promising to realize a higher resolution display based on the bistable ERPC.

A multi-color display could be realized through the precise control of pixels containing bistable ERPCs with different colors. (Fig. 4a–f) It was well known that the structural color of colloidal PC could be tuned by the size of building blocks. Typically, monodisperse SiO$_2$ particles with average diameter of 199 nm, 169 nm, 138 nm and standard deviations of 2.98, 1.46, 1.87 could be used to prepare ERPCs with red, green, and blue colors. (Supplementary Fig. 23) All these ERPCs showed a colored or colorless appearance at 0 V and could be switched to each other by a short-time voltage. (Supplementary Fig. 24) More importantly, the RGB colors from the ERPCs had high saturation and good monochromaticity, which could be confirmed by the narrow reflection peaks at 632 nm, 541 nm, and 475 nm. It was also supported by the coordinates in the CIE chromaticity diagram, in which the points close to the spectral locus corresponded to colors

with high saturation. Such high color saturation and brightness were exactly the advantage of the bistable ERPC compared to the traditional electrophoretic color display based on the color-filter and black-white E-inks. (Supplementary Fig. 25).

Based on the high saturation and good monochromaticity, the three bistable ERPCs could be used to realize a multi-color display through the spatial mixing of primary colors within a single RGB pixel. (Fig. 4g–j) Here, a prototype display panel composed of the patterned ITO glass, the ERPC array fixed by epoxy frames, and the top ITO glass were fabricated following the procedures in the experimental section. The resultant RGB pixels (8 mm × 8 mm) contained red, green, and blue subpixels, and each of them could be controlled independently by programmed voltages so that more secondary colors could be output through color mixing. For example, when the blue ERPC was switched to the colorless state, the pixel would show an orange color due to the mixing of red and green. This could be verified by the reflection spectra with two individual peaks at 535 nm and 636 nm, and also the coordinates in the CIE diagram exactly located between the coordinates of the primary colors. Similarly, the pixel showed a cyan or magenta color when the red or green ERPC was switched to the colorless state. Limited by the manufacturing crafts, the visual effect of color mixing still needed improvement, and a better effect should be accessible when the size of the RGB pixels was further decreased.

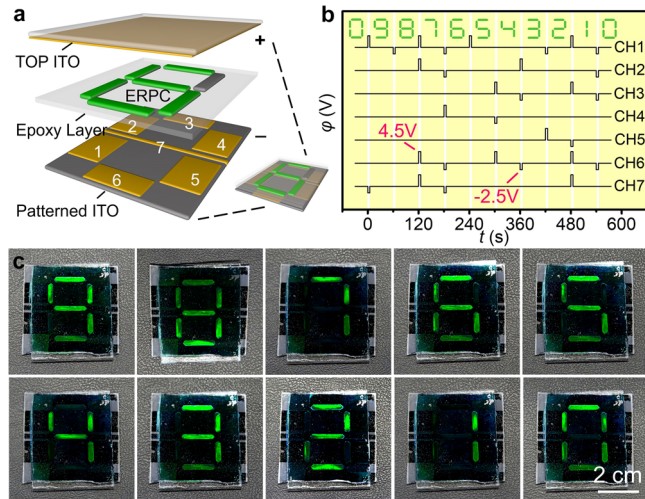

**Fig. 5 | Numerical indicator based on bistable ERPC. a** schematic diagram of the numerical indicator composed of seven parts of independently controlled SiO$_2$/PCb-PEG-EG ERPC (30/60/5/5 vol%); **b** the voltage profiles of Channel 1–7 for the sequent display of ten digits; **c** displaying the digits from 9 to 0 in the absence of E-field.

The greyscale of the bistable ERPC could be achieved by controlling the density of colored subpixels or applying the bias voltage during the recovery of the colored state. Benefiting from the accurate fabrication of patterned ITO electrodes in micron-scale, one can prepare a small green pixel (1.8 mm × 1.8 mm) composed of 4 subpixels. The subpixels could be independently controlled by the different potential applied to the 4 electrodes, and gradually illuminated to realize an increasing in reflection intensity and 4-level of greyscale in green color. (Supplementary Fig. 26) Another method to control the greyscale was the application of bias voltage to suppress the colloidal assembly during the recovery of the colored state. For example, a 1.7 V of bias voltage could be applied at different time to cooperate with the viscous retarding force and balance the interparticle repulsion, which froze the reflection at specific intensity and realized different greyscales. (Supplementary Fig. 27 and 28).

As a demonstration, we fabricated a numerical indicator using green ERPC and exhibited its working details in practical applications. (Fig. 5) As shown in the schematic diagram, the display unit consisted of a patterned ITO glass at the bottom, electrophoretic cells, and an untreated ITO glass at the top. The seven electrophoretic cells that constituted the numerical indicator were independently controlled by voltages programmed in Channel 1 - 7 so that they could function cooperatively to show any digits from 0 to 9. For instance, the displayed digit was changed from "9" to "8" when an inverse voltage of −2.5 V in Channel 1 was applied to switch the ERPC in Cell 1 from the colorless to the colored state. Similarly, the displayed digit could be further changed to "7" when forward voltages of 4.5 V in Channel 1, 2, 6, and 7 were applied to switch the ERPCs from the colored to the colorless state. Short movies were recorded to show the switching between "8", "3", and "6". (Supplementary Movie 1, 2) During the working of the display unit, the external voltage was only required when refreshing the digits, and it was not required to maintain the display.

## Discussion

In summary, a SiO$_2$/PCb-PEG-EG ERPC with bistable and switchable characteristics was developed for electrophoretic color display. The resultant ERPC possessed two stable states at 0 V, which were the colored state with highly ordered particle arrangement and the colorless state with disordered particle stackings. PEG was the critical component, which increased the viscosity, froze the Brownian motion of particles, and stabilized the disordered structure. These two states could be reversibly switched by the exertion of a forward voltage at 4.5 V for 5 s and an inverse voltage at −2.5 V for 4 s. In this way, an ERPC based on "colloidal assembling and disassembling" but not "lattice shrinkage and expansion" was achieved, and its bistable property addressed a big concern for ERPC to be used as electrophoretic inks. The volume fraction of PEG, EG, and SiO$_2$ in ERPC, as well as the waveform of the E-field, were optimized to realize the long-time holding of the colorless state and the quick recovery of the colored state. With optimized conditions, the bistable display and reversible switching could be realized in most working scenarios. The as-prepared ERPC had the potential to realize the high-resolution display because the colloidal microcrystals usually had a size from several to tens of micrometers, and its optical signal was not affected unless below such size. The ERPC also had the potential for a multi-color and greyscale display, where more colors could be produced through the spatial mixing of primary colors within a single RGB pixel, and the greyscale could be controlled by the density of colored subpixels or the bias voltage applied during the recovery of the colored state. Finally, a numerical indicator based on bistable ERPC was fabricated to confirm its advantage in the low-power display.

Currently, a slow recovery of the colored state is clearly the main limitation for the bistable ERPC based on the assembling-disassembling mechanism. Unlike the traditional ERPC with fast color tuning related to the slight change of crystal lattice, the switching from the colorless to colored state for bistable ERPC is intrinsically a crystal growth process, and the addition of viscous component further slows down this process. Based on these understandings, people may find possible methods to accelerate the response in the future works by tuning the ERPC's ingredients and the E-field waveforms. For example, a decrease of $f_{PEG}$ and an increase of $f_{SiO2}$ in ERPC are both favorable to speed up the reassembly of colloidal crystals. (Supplementary Fig. 29) A slightly higher working temperature and specific E-field waveforms, such as impulse or step-form modulation, also help to shorten the recovery time. (Supplementary Fig. 30) With integration of optimized conditions, the recovery of the colored state can be achieved in 22 s. Furthermore, the coupling of fixed RGB PC layer with optical shutters having fast switching ability can be another technical route to realize bistable PC display with significantly improved response speed, where the simultaneous manipulation of color and greyscale can display any color images.

It has to be admitted that there is still a long way for the bistable ERPC to be applied in the commercial display. Except for the bistable and switchable optical signals, a mature electrophoretic display technology also has to meet the requirements in color and greyscale, resolution, refresh rate, viewing angle, power consumption, and service life. With continuous efforts to optimize the recipe and E-field modulation, bistable ERPC is believed to have applications in the energy-saving display devices.

## Methods
### Materials
Tetraethyl orthosilicate (TEOS, 98%), aqueous ammonia (NH$_3$ • H$_2$O, 28%), ethanol (EtOH, 99%), and polyethylene glycol (PEG, Mw = 1500 Da, 10000 Da, 20000 Da) were obtained from Sinopharm Chemical Reagent Co. Ltd. Ethylene glycol (EG, 99%) was purchased from J&K Co. Ltd. Propylene carbonate (PCb, 99%) and arginine (98%) were obtained from Aladdin Co. Ltd. Indium tin oxide (ITO) coated glasses (sheet resistance ≤5 Ω/□) were purchased from South China Xiang Science & Technology. All chemicals were used without further purification.

### Synthesis of SiO$_2$ particles
Monodispersed SiO$_2$ particles were prepared by a seeding growth method. First, the SiO$_2$ nanoparticle seeds were prepared by hydrolysis

of TEOS (5.5 mL) in water (87 mL) with arginine (0.087 g) as surfactants for 24 h. The reaction solution was cooled down and stored as seed solution for further use. In a typical synthesis of $SiO_2$ particles with a diameter of 138 nm, a seed solution (250 µL) was mixed with EtOH (200 mL), $H_2O$ (14 mL), and $NH_3 \cdot H_2O$ (8 mL) at room temperature, after which TEOS (16 mL) was gradually injected at a speed of 4 mL/h to produce $SiO_2$ particles in 12 h of reaction. $SiO_2$ particles with diameters of 169 nm or 199 nm were prepared by similar procedures, except that 190 µL or 120 µL of the seed solution was used at the beginning. The particles were collected by centrifugation, washed with ethanol three times, and finally dispersed in ethanol.

### Preparation of bistable ERPC
According to the previous report on the synthesis of liquid PC, the $SiO_2$ particles (0.030 $cm^3$) were first dispersed in the mixing solution of EtOH (1000 µL), PCb (60 µL), EG (5 µL), and PEG-20000 (6.35 mg, 0.005 $cm^3$) to form a homogeneous colloidal suspension. It was then heated at 90 °C for 2 h to evaporate the excessive EtOH and cooled down to room temperature to produce the ERPC (100 µL). The ERPC was simply sandwiched between two ITO glasses with an interspacing of 50 µm or encapsulated in the designed display unit for the test of electrical responses.

### Fabrication of electrophoretic display unit
The typical PC electrophoretic display unit had a triple-layer structure, including a patterned ITO glass at the bottom, an epoxy adhesion layer with patterned cells in the middle, and a raw ITO glass at the top. Firstly, a double-sided adhesive tape was engraved by the laser engraver and attached to the top raw ITO glass to construct patterned grooves to contain the ERPCs. Secondly, the conductive layer of the bottom ITO glass was selectively removed by laser engraving or chemical etching to form a patterned electrode, which would be aligned to the epoxy grooves to construct the complete electrophoretic cells. Finally, two ITO glasses were aligned and adhered to each other after the addition of ERPCs and sealed by optical adhesives on the edges to produce prototype display units. The contacts on the patterned ITO electrode were connected to different channels of an external DC power source so that the electric field and the structural color of each unit cell could be independently controlled. As a light absorber, a black 3 M insulation tape was attached to the bottom ITO glass so that the display unit showed black appearance when the bistable ERPC was in a colorless state.

### Measurement of the time-resolved reflection spectra
The dynamic time-resolved reflection spectra were continuously collected by an Ocean Optics Maya 2000 Pro spectrometer. The fiber probe was held steadily above the PC electrophoretic display unit in a perpendicular way. Before the programmed voltage was applied to the display unit, the spectrometer was already started to record the reflection spectra in a step of 1 sec so that the reflection changes of the bistable ERPC during the entire switching could be recorded. The measured data could be plotted in a contour map of dynamic reflection spectra with time ($t$) on the $x$ axis, reflection wavelength ($\lambda$) on the $y$ axis, and reflection intensity ($R$) in color. They could also be plotted as an "$R$-$t$" curve, which was used to indicate the color switching for ERPCs or confirm their bistable characteristics.

### Characterizations
The morphology and particle size of $SiO_2$ particles were characterized by an FEI Tecnai G2 F30 transmission electron microscope (TEM). The kinematic viscosities of the mixing solvents were measured by a Puruite PND401a intelligent kinematic viscosity testers. The photos of the ERPCs were taken with a Fujifilm X-A3 digital camera. The optical microscopic (OM) images were collected by an Olympus BXFM reflection-type microscope operated in dark-field mode. The self-

assembly structure of $SiO_2$ particles in ERPC was analyzed by the Hitachi S4800 scanning electron microscope (SEM) after freeze-drying the ERPCs. The reflection and transmission spectra of ERPC were collected by an Ocean Optics Maya 2000 Pro spectrometer, while the microscopic reflection spectra were measured by Idea optics NOVA-EX coupled to the optical microscope. The reflection spectra and the color saturation for the color-filter and the bistable ERPC electrophoretic display were measured by a 3NH YS3020 colorimeter equipped with an integrating sphere with a size of 48 mm. The external electric field with programmed profiles was supplied by a PR35-5A-3CP DC power source.

### Reporting summary
Further information on research design is available in the Nature Portfolio Reporting Summary linked to this article.

## Data availability
The data that support the findings of this study are available within the paper and Supplementary Information. Additional relevant data are available from the corresponding author on request.

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

## Acknowledgements

J.P. Ge thanks the funds from the National Natural Science Foundation of China (21972046, 22172054).

## Author contributions

J.P. Ge and Q.Q. Fu conceived the idea, designed the research, and wrote the manuscript. Q.Q. Fu performed the ERPC studies and conducted the experiments. W.Y. Yu and G.Y. Bao contributed to data collection and analysis. J.P. Ge supervised and managed the research work. All authors discussed the results and contributed to the writing and revision of the manuscript.

## Competing interests

The authors declare no competing interests.
