## [Peer Review File · Nature Communications]

Reviewer comments, first round-

Reviewer #1 (Remarks to the Author):

This manuscript is an intriguing advance in photonic crystal-based, electrically tunable devices. The authors describe a device architecture whereby a colloidal dispersion can be electrically switched from an ordered colloidal arrangement showing colour reflection, to a disordered colourless state. Importantly, there seems to be bistability of both these states.

This work will be of significance to this field of study. As the authors note, the achievement of bistability is an important achievement, and one that has not yet been demonstrated by an electrophoretic-based system (although bistability has been demonstrated in other electrochemical systems, such as in citations 13-15 in the current manuscript). As such, it should be published in a peer-reviewed journal. However, I do not believe it merits publication in a prestigious journal such as Nature Communications.

The authors describe this work as a significant advance in the field of electronic paper. However, there are some important drawbacks to this system which would reduce its impact on this field:

- This looks to be a material which is only capable to ON-OFF switching between a single color and a colorless state. While multi-colour displays are possible, this is only so by preparing different liquid formulations and having these as side-by-side regions in sub-pixels. There is no benefit to this approach versus putting colour filters over a commercial electrophoretic display.
- Furthermore, there seems to be no possibility of achieving greyscale control of the peak intensity. Once a voltage pulse is applied to switch to the coloured state, the system seems to spontaneously assemble into its equilibrium state (most ordered) over time. Without the possibility of greyscale control (which commercial electrophoretic displays do well), this technology wouldn't have any real commercial potential.
- Because the mechanism is highly dependent on the mixture viscosity, it would be expected to show greatly varying performance at different ambient temperatures. This would greatly limit any real-world application of the technology.

For the most part, the work and analysis in this manuscript is sound, and there is sufficient detail provided for the work to be reproduced. However, there are some conclusions/claims that are lacking sufficient experimental evidence or support:

- In the manuscript, the authors claim that the tuning effect they are seeing is due to the electrophoretic movement of particles and their freezing in place due to locally increased viscosity. This may well be the correct mechanism. However, in the devices they have made the liquid colloidal crystal dispersions are in direct contact with the electrodes. At the voltages used (up to 4.5V), there could be electrochemical processes taking place such as electrolysis or electrochemical breakdown of some of the components of the liquid mixture; In particular species with active hydrogens such as EG, PEG (-OH end groups), arginine (used as surfactant in the synthesis of silica particles), or adventitious water as there was no mention of excluding moisture from some of the hygroscopic components (EG, PEG, silica). In order to support their proposed mechanisms, the authors should run an experiment where they exclude the possibility of electrochemical reactions by passivating one or both electrode surfaces with a thin layer of electrical insulator – for instance, a layer of cured epoxy resin or other crosslinked polymer could be deposited by spin-coating from a dilute solvent solution, followed by curing. Provided the layers of electrical insulator are thin, they should not substantially affect the electric field across the 50 micron cell gap. If the authors see the same type of electrical response it would support their proposed mechanism; if not their mechanism would not be supported and any conclusions related to long-term device stability and compatibility with electric-field based device architecture would have to be revised. This same type of experiment was carried out in other publications – For instance in cited reference 16, the authors passivated the electrode to confirm their proposed mechanism was indeed based on electrochemical actuation, and not a field effect. Furthermore, the authors should measure current flow during the application of tuning voltage to ensure it is not

higher than expected for purely capacitive charging.

- To further support their proposed mechanism of bistability achieved through local viscosity increase, the authors should carry out tuning experiments at temperature higher than room temperature. An increased temperature will cause viscosity decreases, and should reduce the bistability as well as the time required to revert to the reflective coloured state.

- The authors state (lines 187-189) "Similar to the introduction of PEG, an excessive fraction of EG, such as 10%, was unfavorable to the recovery of the colored state due to high viscosity, so 5% of EG was probably the optimized condition". This is doubtful. While polymers are known to have dramatic effects on viscosity, and an increase from 5% to 10% can bring about a very sizeable increase, small molecules do not have this effect. For example, in mixtures of water and ethylene glycol (<https://www.meglobal.biz/wp-content/uploads/2019/01/Monoethylene-Glycol-MEG-Technical-Product-Brochure-PDF.pdf>), an EG content of 10% vs 5% only gives a marginally different viscosity. While I do not have access to data for propylene carbonate/ethylene glycol mixtures, the authors could easily make this measurement themselves. If this measurement does not support their claim (as I suspect), the authors would need a different explanation for why this increased EG content leads to such a deviation in electrical response.

Reviewer #2 (Remarks to the Author):

Photonic Crystals are promising candidates for color display. In this manuscript, the authors presented ERPC with bistable and switchable characteristics based on "colloidal assembling and disassembling", and developed a new color E-ink for reflective display. Furthermore, the potential of PC E-ink for high resolution and multi-color display is also investigated by prototype display panels. This is an important study to paved a way for ERPC as next generation color E-ink, and certainly will bring interets to researchers in multiple fields. The following questions should be considered to improve the manuscript.

1. The word "bistable" is usually used for the state of a structure. For the ordered and disordered stacking states in this study, the authors are suggested to use another word.
2. Electric stimuli usually lead to quick response (hopefully, on the order of ms). However, in this study, the response time is on the order of s. For display, a quick response is also necessary. It's better to add discussions on how to improve the response time.

Reviewer #3 (Remarks to the Author):

The Authors presents a new recipe for an electronic-ink (E-ink) based on electrically-responsive photonic crystals (ERPCs) working through a "colloidal assembling and disassembling" principle. The optimization of the latter mechanism allows them to obtain a reversible, bistable, switchable color characteristic for their ERPC E-ink recipe.

Importantly, the Authors' ERPC E-ink is able to provide stable two-state operation even in the absence of electric stimulus. This property represents a great step forward for the development of electrophoretic color E-ink, which, up to date, were only limited to work under an applied electric field. Moreover, the Authors showed a color E-ink display prototype demonstrating the high potential of the proposed solution.

This work is clearly highly significant in the specific field of ERPC, and also appealing for a broad audience. The manuscript is clear and accurate, and the claimed results are well supported by experimental data, and by the extensively reported supplementary materials.

From both scientific and technological points of view the manuscript is really sounding.

For these reasons, I consider the Authors' work worthy of publication in this Journal after considering the following minor suggestions/revisions.

- Figure 2: for each of the three investigations performed on ingredient concentrations and molecular weight, the fixed concentration for the complementary ERPC components should be

reported for the sake of clarity. In this way, the Reader will better understand the evolution of E-ink formulation during recipe optimization (ex. Fig 1a-b-c were obtained with ??% of SiO₂, ??% of PCb and ??% of EG).

- For what regards the optimization of the color switchable performance on the electric field waveform and modulation, it would be interesting to more deeply investigate the mechanism behind this. As the main limitation of the shown demonstrator is the time response, the Authors should at least better discuss how the performance increase depending from the electric field modulation. Should this optimization be dependent on the used recipe? In other words, the final shown performance is the ultimately optimized one?
- In the discussion, the Authors should better highlight the main limitations of their E-ink recipe and possible perspective for improvement.
- About SiO₂ particles, Authors provide very precise diameters (199 nm, 169 nm, and 138 nm), but it is not reported how monodispersed are that particles, neither an error is specified. Has been diameter dispersion characterized for the different types on nanoparticles? If not, it should be done (by TEM or by z-sizer).

Reviewer #4 (Remarks to the Author):

This manuscript deals with the first report of a tunable photonic crystal device with bistability using colloidal assembling/disassembling nature through electric field manipulation. Therefore, the novelty of the manuscript is publishable. Without a doubt, bistability is essential for reflective display in terms of saving energy consumption, which is one of the advantageous features of reflective displays like e-books compared to conventional transmissive displays such as LCD and OLED. Although this methodology provided bistability, It is not sure this methodology is more promising than one with a patterned RGB photonic crystal with an optical shutter. Also, this methodology loses one crucial feature of tunable photonic crystal: the ability to develop a one-pixel full-color device, which can confer three times brighter images than a display with RGB pixels. Therefore, the revised manuscript should include a proper response to the following questions.

1. It seems like introducing 5% of EG not only increased recovery to the assembling state but also enhanced the holding disassembling state (compare Fig 2 (b) and (d)). The authors should provide a proper explanation for this. Also, what happens if the system excludes PEG and replaces it with EG ?
2. How long can the disassembling state be held without applying an electric field? It seems like the reflectivity in the off state (disassembled state) keeps increasing before applying the negative field (Supplementary Fig. 10).
- 3 Why not pattern photonic crystal layer (R/G/B) with high reflectivity and integrate another layer with an optical shutter with fast switching nature having bistability because the suggested system exhibits slow response and low reflectivity. Please discuss this matter in the revised manuscript.
4. The "Discussion" section is like a summary or conclusion in this manuscript.

Reply to Reviewer #1:

- This looks to be a material which is only capable to ON-OFF switching between a single color and a colorless state. While multi-colour displays are possible, this is only so by preparing different liquid formulations and having these as side-by-side regions in sub-pixels. There is no benefit to this approach versus putting colour filters over a commercial electrophoretic display.*

--- Liquid crystal is a responsive material with on-off switching controlled by the arrangement of LC molecules. The simple on-off switching doesn't affect its wide application in display and it will not do so for the bistable ERPC either. We admit that using ERPC liquids with three formulations to realize a multi-color display may bring some difficulty in assembling the display panel and raise its cost compared to the display with single responsive liquids, but such encapsulation can be fulfilled by accurate injection with current advanced manufacturing crafts.

Furthermore, almost all the multi-color display panels, from the CRT to LCD and then LED displays, are composed of side-by-side or other arrangements of RGB sub-pixels. Considering the current work aims at developing "a new material" for color E-ink display based on bistable ERPC rather than proposing "a new design strategy" for the color display panel, it should not affect the novelty when using the broadly accepted design to fabricate the prototype display panel.

As for the advantages of the "bistable ERPC" E-ink display compared to the "color-filter" E-ink display, we made a full comparison from the working mechanism to the practical display effects to demonstrate that the "color-filter" E-ink display has unsatisfactory brightness and color saturation. As shown in the following figures, a commercial E-book reader using "color-filter" technology is compared with the color E-ink display based on bistable ERPC. Both visual observation and the measurement of color saturations (C) suggest that the "color-filter" E-ink display has a lower intensity and a lower color saturation compared to the other. This is because the color-filter decrease the intensity of both incident light and reflected light, which makes the reflected RGB light dimmer. Furthermore, the wide range in wavelengths of the transmitted light going through the color filters determines the reflected light has poor monochromaticity, as the white spheres reflect lights with all wavelengths. Therefore, the reflection peaks from the "color-filter" E-ink display are usually much broader and weaker than those from the PC display. In other words, the "bistable ERPC" E-ink technology has apparent advantages in color brightness and saturation.

We had emphasized the advantage of the “bistable ERPC” E-ink in the discussion of Figure 4 on Page 11, and its comparison to the “color-filter” E-ink display was supplied as Supplementary Figure 25 on Page S31.

Supplementary Figure 25. The “color-filter” vs. the “bistable ERPC” E-ink display. a, b) working mechanism, c, d) reflection signals, e, f) visual appearance, and g) color contrast for the color E-ink display based on color-filter and bistable ERPC.

2. Furthermore, there seems to be no possibility of achieving greyscale control of the peak intensity. Once a voltage pulse is applied to switch to the coloured state, the system seems to spontaneously assemble into its equilibrium state (most ordered) over time. Without the possibility of greyscale control (which commercial electrophoretic displays do well), this technology wouldn't have any real commercial potential.

--- We proposed two possible solutions to address the concern of greyscale control. Firstly, based on the advanced manufacturing crafts, the greyscale of PC E-ink can be controlled by the density of

“colored” subpixels. As shown below, we had prepared a small green pixel (1.8 mm×1.8 mm) composed of 4 subpixels, whose E-fields were independently controlled by 4 electrodes separated by chemical etching. Digital photos, optical microscope images, and corresponding reflection spectra all suggested that a 4-level of greyscale could be realized by gradually turning the subpixels into colored state. Considering the commercial “black/white” E-ink display usually has a greyscale level of 4 to 16, the aforementioned independent control and coupling of subpixel’s signals can be a reliable way to achieve greyscale control for the current PC E-ink.

Supplementary Figure 26. Greyscale controlled by subpixel density. a) Digital photos, b) optical microscope images, and c) reflection spectra of green PC E-ink with 4 different greyscales controlled by the density of colored subpixels.

Another method to control the greyscale of PC E-ink is the application of “bias voltage” during the recovery of reflection intensity. As shown below, after 4.5 V was applied to the bistable ERPC for 1s (not 4s), it quickly recovered to the strongest reflection intensity even without the exertion of inverse voltage, because 1s of 4.5V was not long enough to achieve stable disordered state. When the reflection intensity was recovered to a specific value, the application of 1.7 V of bias voltage could cooperate with the viscous retarding force to balance the interparticle repulsion, which froze further assembly of colloidal PCs. In this way, the reflection intensity could be maintained, and the intensity could be controlled by the time to apply the bias voltage. Following the above strategy, red, green, and blue PC E-ink with 4 different greyscales were achieved.

Supplementary Figure 27. Reflection intensity maintained by the bias voltage. a, d) Dynamic reflection spectra (DRS) and E-field modulation for a green PC E-ink showing the gradual recovery of reflection intensity after 4.5V was applied for 1s. b, e) A specific reflection intensity can be maintained when a bias voltage of 1.7 V is applied during the recovery process; c, f) 3 kinds of reflection intensities realized by the programmed application of bias voltage.

Supplementary Figure 28. Greyscale controlled by reflection intensity. a-c) Digital photos and d) reflection spectra of red, green, and blue PC E-ink with 4 different greyscales controlled by the reflection intensity.

In summary, we developed two kinds of greyscale control for PC E-ink. The first strategy is feasible and reliable, but the second strategy still needs to be improved as it requires a continuous application of bias voltage. Although 1.7V is a relatively low potential and consumes much little electrical power than most displays, we still wish to find methods to maintain the reflection intensity

at 0V in the future studies. We had supplied the discussion of “greyscale” to the manuscript on Page 12, and related results were supplied as Supplementary Figure 26-28 on Page S32-S34.

- 3. Because the mechanism is highly dependent on the mixture viscosity, it would be expected to show greatly varying performance at different ambient temperatures. This would greatly limit any real-world application of the technology.*

--- The working of bistable ERPC is highly dependent on the viscosity because the switching between the colored and colorless state is actually switching between a concentrated colloidal system ($f_{\text{SiO}_2} = 30\%$) and a more densely packed colloidal system ($f_{\text{SiO}_2} = 50\%$). The relatively low viscosity of the former system allows the spontaneous colloidal assembly to form the colored state, and the much higher viscosity of the latter system freezes the Brownian motion to hold the disordered state. As requested by the reviewer, we had investigated the temperature influence on ERPC’s performance as the viscosity usually decreased along with the rise of temperature. In the experiment, we placed the SiO₂/PCb-PEG-EG ERPC in an incubator with temperature precisely controlled at 10 °C, 25 °C, 40 °C, 50 °C, and 60 °C and tested the time evolution of reflection intensity (R-t) as the ERPC was switched between the ordered and disordered state. The results were used to determine whether the ERPC still possessed the bistable and switchable characteristics.

At temperature below room temperature ($T = 10^\circ\text{C}$), the colorless state was well kept after a forward voltage was removed, but it would take a longer time to recover to the colored state, both of which were attributed to the increased viscosity at low temperature. When the temperature was around or higher than the room temperature ($25^\circ\text{C} < T < 60^\circ\text{C}$), the disordered state could also be held for 10 min, probably because the viscosity was majorly dominated by the high volume fraction of SiO₂ and PEG but not temperature, and the viscosity was still high enough to freeze the particles’ Brownian motion. Certainly, the decreasing of viscosity along with the rise of temperature was detectable, as the reflection intensity recovered faster at higher temperatures no matter in the colorless state or during the recovery towards the colored state. Further increasing the temperature was not recommended for the current material system because it not only brought challenges to the holding of colorless state but also made higher demands on the encapsulation of ERPC to avoid evaporation of solvent and dissolution of adhesives.

Based on the above experiments, one can conclude that the temperature fluctuation in a reasonable range around the room temperature has little influence upon the bistability and switching performance of PC E-ink. It will not limit the practical application of current technology. A brief discussion about the temperature influence was supplied to the manuscript on Page 6, and the related results were supplied as Supplementary Figure 3 on Page S9.

Supplementary Figure 3. Temperature influence on the bistable characteristics. The change of reflection spectra of SiO₂/PCb-PEG-EG ERPCs at a-b) 10 °C, c-d) 25 °C, e-f) 40 °C, g-h) 50 °C, i-j) 60 °C and the corresponding time evolution of reflection intensities when k) 4.5V was applied to the ERPC for 5s to turn it from the colored state to the colorless state, and -2.5V was applied to the ERPC for 4s to turn it from the colorless state to the colored state.

4. *In the manuscript, the authors claim that the tuning effect they are seeing is due to the electrophoretic movement of particles and their freezing in place due to locally increased viscosity. This may well be the correct mechanism. However, in the devices they have made the liquid colloidal crystal dispersions are in direct contact with the electrodes. At the voltages used (up to 4.5V), there could be electrochemical processes taking place such as electrolysis or electrochemical breakdown of some of*

the components of the liquid mixture; In particular species with active hydrogens such as EG, PEG (-OH end groups), arginine (used as surfactant in the synthesis of silica particles), or adventitious water as there was no mention of excluding moisture from some of the hygroscopic components (EG, PEG, silica). In order to support their proposed mechanisms, the authors should run an experiment where they exclude the possibility of electrochemical reactions by passivating one or both electrode surfaces with a thin layer of electrical insulator – for instance, a layer of cured epoxy resin or other crosslinked polymer could be deposited by spin-coating from a dilute solvent solution, followed by curing. Provided the layers of electrical insulator are thin, they should not substantially affect the electric field across the 50 micron cell gap. If the authors see the same type of electrical response it would support their proposed mechanism; if not their mechanism would not be supported and any conclusions related to long-term device stability and compatibility with electric-field based device architecture would have to be revised. This same type of experiment was carried out in other publications – For instance in cited reference 16, the authors passivated the electrode to confirm their proposed mechanism was indeed based on electrochemical actuation, and not a field effect. Furthermore, the authors should measure current flow during the application of tuning voltage to ensure it is not higher than expected for purely capacitive charging.

--- As requested, we ran a passivation experiment to demonstrate the “electrophoretic mechanism” of the bistable ERPC system. As shown in the following figure, the original ITO glass had a very pale purple color. When the ITO glass was covered by a thin layer of cured optical adhesive (Norland optical adhesive, NOA 61), it became colorless and transparent. The cross-sectional SEM image confirmed the successful deposition of the NOA61 layer on the ITO glass. Meanwhile, the top-view SEM image proved the NOA61 layer was insulated, as an insulated object in SEM observation was usually much darker than the conductive objects. Then, we used the passivated ITO glass as one electrode to fabricate the electrophoretic cell, and the as-prepared ERPC still showed the same electrical response as presented in the original submission. This passivation experiment proves that the bistable ERPC works by the “electrophoretic” mechanism rather than the “electrochemical” mechanism.

Furthermore, the current flow during the application of external voltages were measured as below. A statement denying the “electrochemical mechanism” was supplied to the manuscript on Page 6, and the results of passivation experiment were supplied as Supplementary Figure 4 on Page S10.

Supplementary Figure 4. Electrophoretic mechanism proved by the electrode passivation. a) digital photos, b) cross-sectional SEM image, and c) top-view SEM image of NOA61-coated ITO glass. d, e) the colored and the colorless state of the bistable ERPC sealed between a NOA61-coated ITO and an untreated ITO electrode.

Current flow measured during the application of a) a forward voltage of 4.5 V for 5 s and b) an inverse voltage of -2.5 V for 4 s

5. To further support their proposed mechanism of bistability achieved through local viscosity increase, the authors should carry out tuning experiments at temperature higher than room temperature. An

increased temperature will cause viscosity decreases, and should reduce the bistability as well as the time required to revert to the reflective coloured state.

--- As requested, we performed the switching experiments at temperatures higher than the room temperature. Detailed results and discussions about the temperature influence on bistable characteristics of ERPC could be found in reply to R1Q3. In short word, a less stable colorless state and a faster recovery to the colored state were observed at 60 °C. These results were consistent with the expectation of decreased viscosity at high temperatures, which made it difficult to hold the disordered state but easy to revert to the ordered state. Therefore, they well supported the mechanism of bistability achieved through local viscosity increase. Fortunately, the temperature influence was little, and the ERPC still possessed the bistability and switching abilities.

6. *The authors state (lines 187-189) “Similar to the introduction of PEG, an excessive fraction of EG, such as 10%, was unfavorable to the recovery of the colored state due to high viscosity, so 5% of EG was probably the optimized condition”. This is doubtful. While polymers are known to have dramatic effects on viscosity, and an increase from 5% to 10% can bring about a very sizeable increase, small molecules do not have this effect. For example, in mixtures of water and ethylene glycol (<https://www.meglobal.biz/wp-content/uploads/2019/01/Monoethylene-Glycol-MEG-Technical-Product-Brochure-PDF.pdf>), an EG content of 10% vs 5% only gives a marginally different viscosity. While I do not have access to data for propylene carbonate/ethylene glycol mixtures, the authors could easily make this measurement themselves. If this measurement does not support their claim (as I suspect), the authors would need a different explanation for why this increased EG content leads to such a deviation in electrical response.*

--- In order to explain the inhibited recovery of the ordered state by excessive EG, we performed a complete investigation of the viscosities of PCb-EG solutions, PCb-PEG-EG solutions, and the SiO₂/PCb-PEG-EG colloidal solutions with f_{EG} changing from 0% to 10%. Here, PCb is the major component of the solution, whose fraction changed accordingly with the others. f_{PEG} is fixed at 5% in the last two solutions, and f_{SiO_2} is fixed at 30% in the colloidal solution.

As expected by the reviewer, a gradual replacement of PCb by EG from 0% to 10% will not cause a large increase in viscosity as the introduction of PEG did, which can be confirmed by the viscosity

of the “PCb-EG” solutions increasing from 2.80 to 3.06 cSt. However, in the presence of PEG (5%), the viscosity of the “PCb-PEG-EG” solution increased from 15.2 to 17.1 cSt as f_{EG} was raised to 10%. In the coexistence of PEG (5%) and SiO₂ (30%) particles, an increased f_{EG} in the “SiO₂-30%/PCb-PEG-EG” colloidal solution caused a significant enhancement of the viscosity from 143 to 186 cSt. Therefore, the viscosity enhancement effect of EG is amplified along with the increasing of total viscosity of the system.

Supplementary Figure 9. Viscosity evolution with f_{EG} . a-c) viscosities of PCb-EG, PCb-PEG-EG, and SiO₂-30%/PCb-PEG-EG solutions measured by the viscometer, and d) viscosity of SiO₂-50%/PCb-PEG-EG solutions calculated from that of SiO₂-30%/PCb-PEG-EG solutions

According to the working mechanism of bistable ERPC, the switching between the colored and colorless state was actually a switching between a concentrated colloidal system ($f_{SiO_2} = 30\%$) and a more densely packed colloidal system ($f_{SiO_2} = 50\%$). The relatively low viscosity of the former system allowed the spontaneous colloidal assembly to form the colored state, and the much higher viscosity of the latter system froze the Brownian motion to hold the disordered state. Since the viscosity of the colloidal system in the disordered state ($f_{SiO_2} = 50\%$) was hard to be measured directly, we had to calculate its value according to the theoretical equation and the viscosity of a less concentrated colloidal solution. “Supplementary Discussion of Viscosity Calculation” indicated the viscosity of

the “SiO₂-50%/PCb-PEG-EG” solution was 4.407 times as that of the the “SiO₂-30%/PCb-PEG-EG” solution. The calculated viscosity of the “SiO₂-50%/PCb-PEG-EG” solution increased from 591 to 771 cSt when f_{EG} was increased from 0% to 10%.

Therefore, the introduction of EG does significantly increase the viscosity of the colloidal system when f_{SiO_2} is 50%. A proper amount of EG helps to enhance the holding of the disordered state due to increased viscosity, but an excessive EG will be unfavorable to the recovery of the colored state also due to the further increased viscosity. These statements are also well supported by the experimental results in Figure 2d and 2e.

To clarify the influence of f_{EG} on viscosity and avoid possible misleading, we had revised the discussion of Figure 2d-2e on Page 7, and the viscosity measurements were supplied as Supplementary Figure 9 on Page S15.

Reply to Reviewer #2:

1. *The word "bistable" is usually used for the state of a structure. For the ordered and disordered stacking states in this study, the authors are suggested to use another word.*

--- The term “bistable” is usually used for the state of a structure. In this work, it refers to the ordered or disordered state of the colloidal particle assemblies in its solution because these two states are stable states which require no external electric field to maintain themselves. Although the term “bistable” is no better than “ordered or disordered” which reveals the physical nature of these two states directly, we still believe it is the most suitable description that we can find. It helps to remind the readers about the connection between the current PC E-ink and the traditional electrophoretic E-ink, where “bistable” or “bistability” are frequently used to define the properties required by the E-ink. (Nature, 1998, 394, 253-255) Through this connection, the term “bistable” may help to attract more attention from academics and industries to further develop such materials in the future.

Limited by our knowledge background, we are unable to provide a more suitable term currently. Any other terms better than “bistable” are welcome to be suggested.

2. *Electric stimuli usually lead to quick response (hopefully, on the order of ms). However, in this study, the response time is on the order of s. For display, a quick response is also necessary. It's better to add discussions on how to improve the response time.*

--- As requested, we had supplied the possible methods to improve the response speed in the section of “Discussion”. The long response time of bistable ERPC, especially for the recovery of the colored state, originated from the “assembling-disassembling” mechanism. Unlike the traditional ERPC with fast color tuning related to the slight change of crystal lattice, the switching from the colorless to colored state for bistable ERPC is more like a “crystal growth” process, and the viscous component further slow down this process. Based on the above understandings, we made some preliminary investigations to shorten the response time by adjusting the E-ink recipe and the E-field waveforms.

Firstly, the volume fraction of PEG was decreased based on the original recipe of SiO₂/PCb-PEG-EG ERPC (30-60-5-5%) to accelerate the recovery of colored state. When the volume fraction of PEG decreased from 5% to 3%, the recovery time reduced from 38 s to 29 s under application of -2.5V for 4s. This is because the reduction of viscosity decreases the viscous retarding force, which facilitates the reassembly of particles. A certain amount of PEG addition ($\geq 3\%$) is necessary for the stabilization of particles and maintaining of the colorless state.

Supplementary Figure 29. Accelerating the recovery of colored state by tuning the ingredients of ERPC. a, b) Influence of volume fraction of PEG-20000 on the holding of “colorless state” and recovery of “colored state”, indicated by the time evolution of reflection intensities; c, d) influence of volume fraction of SiO₂ particles on bistability.

Secondly, the volume fraction of SiO₂ was increased to speed up the recovery of colored state. When the f_{SiO_2} increased from 30% to 33%, the recovery time was shortened from 38 s to 27 s under application of -2.5V for 4s, because the particles tended to precipitate out to form colloidal crystals at higher concentration. However, further increasing of the f_{SiO_2} would increase the viscosity of the colloidal system and cause a decrease in the recovery rate, so that 33% of SiO₂ was the best recipe for the current PC E-ink.

Thirdly, the recovery of the colored state could be accelerated by an impulsive or step-form modulation of the applied voltages. Here, an electrochemical workstation was used to output the electric field waveforms precisely. As shown below, when an inverse voltage of -2.5 V was impulsively applied for 4 times, including application for 1s and interval for 0.5s, the bistable ERPC recovered to the colored state in 32 s, which was faster than the continuous application of -2.5V for 4s (about 40s). In addition, when a step-form E-field with voltage gradually enhanced from -2.0 V to -3.0 V was applied, the recovery time could also be shortened to 30 s. These E-field modulations were probably more matched with the migration of particles and accelerate the recovery of colloidal crystals. With the optimized ingredients (SiO₂/PCb-PEG-EG, 33/57/5/5) and the step-form inverse E-field, the EPRC could recover to the colored state in 22 s.

Supplementary Figure 30. Accelerating the recovery of colored state by E-field modulation. a) The recovery of SiO₂/PCb-PEG-EG (30/60/5/5) ERPC under b) an impulsive electric field, indicated

by time evolution of reflection intensities. c) The recovery of two SiO₂/PCb-PEG-EG ERPCs (30/60/5/5; 33/57/5/5) under d) a step-form electric field.

Furthermore, the recovery of the colored state might also be accelerated by the temperature considering some electronic devices usually work at a temperature slightly higher than the room temperature. (Reply to R1Q3) All the above preliminary investigations to improve the response speed were supplied to the “Discussion” section on Page 13, and the related results were supplied as Supplementary Figure 29-30 on Page S35-S36.

Reply to Reviewer #3:

1. *Figure 2: for each of the three investigations performed on ingredient concentrations and molecular weight, the fixed concentration for the complementary ERPC components should be reported for the sake of clarity. In this way, the Reader will better understand the evolution of E-ink formulation during recipe optimization (ex. Fig 1a-b-c were obtained with ??% of SiO₂, ??% of PCb and ??% of EG).*

--- The ingredients of all bistable ERPC reported in this work have been provided in Supplementary Table 1 with the original submission. As suggested, we now supplied the formulation of E-inks in all the figure captions for a clear understanding of the results and discussions. Please check the revised captions on Page 18-22.

Supplementary Table 1. The composition of all ERPCs tested in the current work.

ERPC	f_{SiO_2} (%)	f_{PCb} (%)	f_{PEG} (%)	f_{EG} (%)	Test
SiO ₂ /PCb-PEG20000-EG	30	60	5	5	Fig. 1
SiO ₂ /PCb-PEG1500	30	67.5-60	2.5-10	0	Fig. 2a
SiO ₂ /PCb-PEG20000	30	67.5-60	2.5-10	0	Fig. 2b
SiO ₂ /PCb-PEG20000-EG	30	65-55	5	0-10	Fig. 2d, 2e
SiO ₂ /PCb-PEG20000-EG	25-35	65-55	5	5	Fig. 2g-2i
SiO ₂ /PCb-PEG20000-EG	30	60	5	5	Fig. 3-5

2. *For what regards the optimization of the color switchable performance on the electric field waveform and modulation, it would be interesting to more deeply investigate the mechanism behind this. As the main limitation of the shown demonstrator is the time response, the Authors should at least better discuss how the performance increase depending from the electric field modulation. Should this optimization be dependent on the used recipe? In other words, the final shown performance is the ultimately optimized one?*

--- As suggested, we had explained the working mechanism of E-field modulation more deeply in the discussion of Figure 3 with the assistance of Supplementary Figure 14-17. Such E-field modulation to ERPC response is basically irrelevant to the ERPC recipe. Please check the revised manuscript on Page 8-9, which is excerpted as below:

In addition to the recipe of PC E-ink, the electric field was also modulated in the form of specific waveforms to enhance the holding of disordered state and accelerate the recovery of ordered state at the same time. It should be noted that the following E-field modulations usually had a similar influence on PC E-inks with different recipes. Because such modulation functioned through the time-resolved electrophoretic forces, which were seldom affected by the particle size or small change of solvent's volume fractions. In the switching from the colored state to the colorless state, a long-time stabilization of the disordered structure could be achieved by increasing the E-field strength from 4.0V to 5.0V (**Figure 3a, 3d**) or prolonging the E-field duration time from 5s to 20s at 4.5V (Supplementary Figure 14). Under strong electric field conditions, the assembly of particles could be completely disrupted, resulting in structures that were more disordered and more difficult to restore ordered arrangement. However, too high voltages such as 5.0V or too long exertion such as 20s might cause electrochemical reactions in the system, which made it difficult to reassemble the particles. Therefore, an "intermittent E-field modulation" was adopted, where 4.5 V of voltage was applied for 5 s in every 60 s. (**Figure 3b, 3e**) Compared to the result of a one-time application of 4.5V for 20s, the PC E-ink under intermittent modulation could maintain the colorless state very well in 300 s yet recover to the colored state after applying the inverse voltage. (Supplementary Figure 15) The "assembling and disassembling" mechanism of the current ERPC determined that the long-term holding of disordered state and quick recovery of ordered state were contradictory requirements. The "intermittent modulation" actually offered a balanced solution by constructing an "active" disordered

state so that the colorless ERPC always got itself ready to recover to the colored state under inverse voltage.

In the switching from the colorless state to the colored state, an inverse E-field with a more precise control was required to speed up the recovery of the ordered structure. (Figure 3c, 3f) Generally, a stronger inverse voltage or a longer E-field application led to quicker recovery of the colored state because both of them facilitated the redispersion of densely packed SiO₂ particles and accelerated the restoration of homogeneous colloidal dispersion with f_{SiO_2} of 30%. For example, when the inverse voltage was enhanced from -1.5 V to -3.5 V, or when the exertion of -2.5V was extended from 2s to 4s, the recovery time indeed decreased accordingly. (Supplementary Figure 16, 17) However, an excessively strong or long inverse voltage brought negative effects instead, as it would lead to the compressing of the just recovered colloidal crystal towards the opposite electrode. Such inverse compression caused by 4s of -4.5V or 6s of -2.5V led to the decrease of reflection intensity and wavelength compared to original values. Therefore, 4s of -2.5V was a proper inverse voltage that could be used to recover the colored state in 45 s. Moreover, an “impulsive modulation” of the inverse voltage at -2.5 V with application of 1s and interval of 0.5 s could further shorten the recovery time to 31 s, probably because the exerted coulombic force was more favorable to the redispersing of particles and recovery of colloidal crystals.

3. *In the discussion, the Authors should better highlight the main limitations of their E-ink recipe and possible perspective for improvement.*

--- As requested by the reviewer and the editor, the main limitation of PC E-ink, the possible perspective for improvement, and the technical demands for E-ink display were supplied to the “Discussion” section after the summary of the current work. Related preliminary investigation for improving the response speed can be found in reply to R2Q2. Please check the revised “Discussion” on Page 13, which is excerpted as below:

..... Finally, a numerical indicator based on PC E-ink was fabricated to confirm its advantage in the low-power-consumption display.

Currently, a slow recovery of the colored state is clearly the main limitation for the PC E-ink based on the “assembling-disassembling” mechanism. Unlike the traditional ERPC with fast color tuning related to the slight change of crystal lattice, the switching from the colorless to colored state for bistable ERPC is intrinsically a “crystal growth” process, and the addition of viscous component further slow down this process. Based on these understandings, people may find the possible methods to accelerate the response in the future works by tuning the ERPC’s ingredients and the E-field waveforms. For example, a decrease of f_{PEG} and an increase of f_{SiO_2} in ERPC are both favorable to speed up the reassembly of colloidal crystals. (Supplementary Figure 29) Furthermore, a slightly higher working temperature and specific E-field waveforms, such as impulse or step-form modulation, also help to shorten the recovery time. (Supplementary Figure 30) With integration of optimized conditions, the recovery of the colored state can be achieved in 22 s.

It has to be admitted that there is still a long way for the bistable ERPC to be applied in the commercial display. Except for the bistable and switchable optical signals, a mature E-ink technology also has to meet the requirements in color and greyscale, resolution, refresh rate, viewing angle, power consumption, and service life. With continuous efforts to optimize the ERPC recipe and E-field modulation, PC color E-ink is believed to have applications in the energy saving display devices.

4. *About SiO₂ particles, Authors provide very precise diameters (199 nm, 169 nm, and 138 nm), but it is not reported how monodispersed are those particles, neither an error is specified. Has been diameter dispersion characterized for the different types of nanoparticles? If not, it should be done (by TEM or by z-sizer).*

--- The size distribution of the three SiO₂ particles for the red, green, and blue PC E-ink were characterized by the TEM images with low magnifications. As shown in the following figure, their average diameters were measured to be 199.2 nm, 169.3 nm, and 138.4 nm, with standard deviations of 2.98, 1.46, and 1.87, respectively.

The information about the average particle size and size distribution were supplied in the revised manuscript on Page 11, and the characterizations were supplied as Supplementary Figure 23 on Page S29.

Supplementary Figure 23. SiO₂ particles for RGB PC E-inks. TEM images and size distribution of the SiO₂ particles used in the a, d) blue, b, e) green, and c, f) red PC E-inks.

Reply to Reviewer #4:

1. It seems like introducing 5% of EG not only increased recovery to the assembling state but also enhanced the holding disassembling state (compare Fig 2 (b) and (d)). The authors should provide a proper explanation for this. Also, what happens if the system excludes PEG and replaces it with EG?

--- The introduction of EG has two functions to the bistable ERPC. 1) EG accelerated the recovery of the ordered state due to the strong interaction between SiO₂ particles and EG. The Hansen solubility parameters suggested that EG had a much higher hydrogen bond parameter (δ_h) than PCb, which meant a stronger combination with Si-OH groups on the SiO₂ particles. The enhanced solvation effect led to the enhancement of interparticle repulsion and then accelerated the reassembling process.

2) The introduction of EG also enhanced the holding of disordered state due to the increased viscosity, which also explained the inhibited recovery of the ordered state if EG was excessive. (Reply to R1Q6) For the bistable ERPC, the switching between the colored and colorless state was actually a switching between a concentrated colloidal system ($f_{\text{SiO}_2} = 30\%$) and a more densely packed colloidal system ($f_{\text{SiO}_2} = 50\%$). The relatively low viscosity of the former system allowed the

spontaneous colloidal assembly, and the much higher viscosity of the latter system froze the Brownian motion and held the disordered state.

It should be noted that the viscosity of the SiO₂/PCb-PEG-EG solution was determined by all four components. Generally, a gradual replacement of PCb by EG from 0% to 10% would cause a very small increase in viscosity for “PCb-EG” solutions (η : 2.80 \rightarrow 3.06 cSt). However, in the presence of PEG (5%) or the coexistence of PEG (5%) and SiO₂ (30%) particles, a gradual introduction of EG would cause a much larger increase in the viscosity of the “PCb-PEG-EG” solution (η : 15.2 \rightarrow 17.1 cSt) and the “SiO₂/PCb-PEG-EG” colloidal solution (η : 143 \rightarrow 186 cSt). More importantly, the real colloidal system to explain the holding of the colorless state was the very densely packed colloidal systems with f_{SiO_2} of 50%. Their viscosities were calculated to increase from 591 to 771 cSt when f_{EG} was increased from 0% to 10%. Therefore, the introduction of EG does increase the viscosity of the colloidal system, leading to the enhanced holding of disordered state.

Table. Measurement of viscosity of solutions with different f_{EG} . PCb is the major solvent, whose fraction changes accordingly with the others. f_{PEG} and f_{SiO_2} are fixed at 5% and 30% in the solution if it contains PEG and SiO₂ particles.

f_{EG} (%)	$\eta_{\text{PCb-EG}}$ (cSt)	$\eta_{\text{PCb-PEG-EG}}$ (cSt)	$\eta_{\text{SiO}_2/\text{PCb-PEG-EG}}$ (cSt)
0	2.80	15.19	142.96
5	2.91	16.24	160.81
10	3.06	17.11	186.34

3) The “SiO₂/PCb-EG” colloidal system with complete exclusion of PEG did not possess a stable “disordered state” because it was inadequately viscous to freeze the Brownian motion of SiO₂ particles in the disordered state. When the 4.5V voltage was removed, the “SiO₂/PCb-EG” solution completely recovered to the ordered state in 60 seconds no matter f_{EG} was 2.5% or 5%. Even if the f_{EG} was increased to 10%, the colloidal system still could not hold the “disordered state”. Therefore, PEG is essential for the bistability.

Supplementary Figure 10. Unstable disordered state for ERPC exclude PEG. The change of reflection spectra of SiO₂/PCb-EG ERPCs with f_{EG} of a) 2.5%, b) 5%, c) 10%, and the corresponding time evolution of reflection intensities after d) 4.5V was applied to the ERPC for 5s.

In summary, 1) EG enhances the interparticle repulsion and accelerate the recovery of ordered state. 2) On the other hand, EG does increase the viscosity of the colloidal system. A proper amount of EG helps to enhance the holding of the disordered state, but an excessive EG will be unfavorable to the recovery of the colored state. 3) The increasing of viscosity by EG is far less than that by PEG, so PEG is still essential for the bistability. In the revised manuscript, we had explained the function of EG more deeply on Page 7, and the proofs about the necessity of PEG were supplied as Supplementary Figure 10 on Page S16.

2. *How long can the disassembling state be held without applying an electric field? It seems like the reflectivity in the off state (disassembled state) keeps increasing before applying the negative field (Supplementary Fig. 10).*

--- In our reflection measurements, it should be first clarified that a reflectivity less than 3 (but not around 0) corresponds to a disordered state due to the reflection baseline. A reflectivity larger than 5

is caused by the broad and weak reflection peak, which corresponds to a very dim colored state. After the forward voltage (4.5V) is removed, the reflectivity does increase slowly because the strong interparticle repulsion pushes away the neighboring particles from each other to form a less compact colloidal system with a stronger scattering effect which is responsible for the increase of reflectivity.

Without applying any bias voltage after the removal of 4.5V, the retaining time of the disordered state is tightly related to the molecular composition and particle volume fraction of the PC E-inks, as shown in Figure 2. It is also related to the E-field waveform, as shown in Figure 3a-3b. Simply, the retaining time of the disordered state becomes longer if a more viscous E-ink or a stronger external field is applied. For example, a strictly disordered state can retain 2-3 min when a voltage of 4.5 V is applied to the SiO₂/PCb-PEG-EG (30/60/5/5%) ERPC for 5 s. It can retain a longer time, for example 10 min, when a voltage of 5 V is applied to the same ERPC for 5 s. However, we believe it is meaningless to pursue an absolute long retaining time, as a viscous E-ink and strong field usually lead to a very slow recovery of the colored state.

With a bias voltage intermittently applied to the colorless ERPC, that is 5s of 4.5V after every 60s, both the holding of disordered state and the faster recovery of ordered state can be satisfied. In this case, the disordered state can easily hold for 30 min, and it can still be switched to the ordered state by inverse voltage (Supplementary Figure 18c). Such retaining time is more meaningful for the working of PC E-ink.

3. *Why not pattern photonic crystal layer (R/G/B) with high reflectivity and integrate another layer with an optical shutter with fast switching nature having bistability because the suggested system exhibits slow response and low reflectivity. Please discuss this matter in the revised manuscript.*

--- Many thanks for the suggestions to improve the response time. However, the “bilayer” strategy is actually unable to realize the RGB display with bistability. As suggested by the reviewer, one can fabricate a “bilayer” device composed of “a patterned RGB PC layer” and “an optical shutter with fast switching ability and bistable optical states”. 1) If this PC layer has a static pattern composed of fixed RGB PC structures, such “bilayer” device is not a real reflective display because it can only show an unchanged image. Apparently, the display based on bistable ERPC is better as it can display dynamic images output by the media source. 2) If the PC layer itself is a display device showing dynamic RGB patterns, it should also possess “bistable” characteristics. Only in this way the whole

“bilayer” device can become an E-ink display, which could show color or not under 0V. In other words, the use of a bistable ERPC with RGB color is inevitable no matter an optical shutter is integrated or not. Therefore, the response speed is still determined by the slow switching of bistable ERPC, so the “bilayer” strategy is helpless to improve the refresh rate. This also explains why there is no design of color E-ink display based on a regular RGB display and an optical shutter.

4. *The “Discussion” section is like a summary or conclusion in this manuscript.*

--- Similar suggestion about the supplement of the “Discussion” section has been raised by the other reviewer. (see Reply to R3Q3) On **Page 13** of the revised manuscript, the main limitation of PC E-ink, the possible perspective for improvement, and the technical demands for E-ink display were supplied to the “Discussion” section after the summary of the current work.

Reviewer comments, second round-

Reviewer #1 (Remarks to the Author):

I was pleased to read the most recent version of the document and the authors' response to the reviewers comments. I believe the authors have largely addressed the reservations I and other reviewers had with the manuscript, and in doing so have also produced a more robust and impactful study.

In particular, I believe the following points have been well addressed:

- 1) The additional information on means of achieving greyscale control is helpful. The sub-pixel approach does seem like a viable means of achieving a greyscale, although an RGB pixelated array requiring sub-pixelation on each of the colours may suffer from unduly high manufacturing complexity. However, the possibility of greyscale control using electrical drive scheme adjustments is intriguing and promising.
- 2) The additional experiments on performance at different temperatures are convincing, and indicate that such displays might indeed function over relevant temperature ranges.
- 3) The experiment on passivation of the electrode is clear, and serves to convincingly support the tuning mechanism
- 4) The additional experiments/information on the role of EG/PEG and their effect on viscosity adds value to the overall discussion on the tuning mechanism

However, the following points should be addressed with edits to the manuscript, if possible:

- 1) The authors make a comparison between the colour reflectivity of a typical electrophoretic display compared to their novel PC display, and find the latter to have a higher reflectivity. However, if the authors are using specular reflection spectroscopy to measure the two systems, the comparison is erroneous. Typically PCs are measured using specular reflection spectroscopy at a fixed angle (usually normal incidence), since they reflect most of the light at (angle of reflection) = (angle of incidence). Typical electrophoretic displays instead exhibit diffuse/Lambertian reflectance, and incident light is reflected in a diffuse way over a 180 degree hemisphere. Therefore, the authors cannot state their system has higher reflectivity, without having done reflection measurements using a 180 degree integrating sphere. If the authors have already made these measurements, this should be clearly stated. Otherwise, the authors should either perform those measurements, or clearly state the difference between the measurements and omit the panels on Supplementary Figure 25 e), f), and g) which give a misleading impression of the technology comparisons.
- 2) Although this was not stated in my original review, and has no bearing on the scientific merit of the article, I do have an issue with one aspect of the nomenclature used in the manuscript. The authors use the term "E-ink" in this article as a generic term for electrophoretic displays. However, I strongly object to the term "E-ink" as a generic technology descriptor. "E INK" is a trademark by the E Ink Corporation, first filed in 2006. They have several related trademarks, and have successfully prevented other companies using "E Ink" as a generic technology descriptor through legal action. In this case, the authors could use "electrophoretic displays" or "electronic paper" as generic technology terms. I would note that the authors are not solely responsible for using this term, and there are other papers in the recent literature that also use "E-ink" in the same way – however, this is not a valid reason to perpetuate the mistake. Again, as this does not bear on the scientific validity of the work, I will leave it to the editors and the authors to weigh the validity of my argument.

In conclusion, I would now recommend publication of this manuscript, once the above noted points

are addressed.

Reviewer #3 (Remarks to the Author):

The authors correctly addressed all the comments to the previous version of the paper. I think the paper is now worthy to be published.

Reviewer #4 (Remarks to the Author):

I'm afraid I disagree with the author's answer to question #3. Authors claim that if this PC layer has a static pattern composed of fixed RGB PC structures, such a "bilayer" device is not a real reflective display because it can only show an unchanged image. However, that is not true. Simultaneously manipulating color and greyscale expression can provide any images, although the quality of the reflective display is decided by reflectivity (brightness), color gamut, switching speed, pixel resolution, etc. This manuscript has shown a new concept of switching them on- and off-state of ERPC with the definite sacrifice of color tunability.

Regarding reflectivity (brightness) of the reflective display, color-tunable pixels with ERPC can usually provide a much brighter image, especially in expressing R, G, and B primary colors, because this device does not require shutting down any pixels to express any color. In terms of these facts, the authors should correct the sentence starting with "Similar to" (2nd paragraph on page 3) because readers and other researchers will misunderstand that this technology has both color tunability and switchability, which is not true in this manuscript. In addition, using any shutters with PC color pixels can provide more switching speed benefits; on the contrary, the manuscript shows poor switching speed due to solvent viscosity, but I think this manuscript still has strong potential due to the first demonstration of the on-off switching concept and a nice device demonstration of ERPC after proper further revision.

Reply to Reviewer #1:

1. *The authors make a comparison between the colour reflectivity of a typical electrophoretic display compared to their novel PC display, and find the latter to have a higher reflectivity. However, if the authors are using specular reflection spectroscopy to measure the two systems, the comparison is erroneous. Typically PCs are measured using specular reflection spectroscopy at a fixed angle (usually normal incidence), since they reflect most of the light at (angle of reflection) = (angle of incidence). Typical electrophoretic displays instead exhibit diffuse/Lambertian reflectance, and incident light is reflected in a diffuse way over a 180 degree hemisphere. Therefore, the authors cannot state their system has higher reflectivity, without having done reflection measurements using a 180 degree integrating sphere. If the authors have already made these measurements, this should be clearly stated. Otherwise, the authors should either perform those measurements, or clearly state the difference between the measurements and omit the panels on Supplementary Figure 25 e), f), and g) which give a misleading impression of the technology comparisons.*

--- As requested, we made double checks on the conditions of all optical measurements in the current research work. Almost all the reflection spectra of ERPCs were measured by an **Ocean Optics Maya 2000 Pro** spectrometer with incident and reflected angle set as 0° (normal direction). However, the reflection spectra and the color saturation for the “color-filter” and the “bistable ERPC” display in Supplementary Figure 25 were measured by a 3NH YS3020 colorimeter equipped with an integrating sphere with size of 48 mm. The sample was illuminated by a D65 light source with diffused illumination, and the reflected signals were collected by the integration sphere to output the color space parameters as well as the reflection spectra ranging from 400 nm to 700 nm with an interval of 10 nm. The technical specifications for the 3NH YS3020 colorimeter was inserted below to prove the measurement conditions. Therefore, the optical measurement had considered the difference in reflection mode for different electrophoretic displays. The advantage of “bistable ERPC” display in color saturation compared to the “color filter” display is definite.

The optical measurement conditions in technological comparison of two EPDs were supplied to the Supplementary Methods on Page S3. It was also clearly stated in the caption of Supplementary Figure 25 on Page S31.

5.2 Technical Specifications

Optical Geometry	Reflect: di:8 °, de:8 (diffused illumination, 8-degree viewing angle);SCI&SCE / SPIN&SPEX;Including / excluding UV measurements(some models have); Comply to standard CIE No.15, GB/T 3978, GB 2893, GB/T 18833, ISO7724/1, ASTM E1164, DIN5033 Teil7	Wavelength Range	400-700nm
		Wavelength Interval	10nm
		Semiband Width	10nm
		Measured Reflectance Range	0-200%
Features	high accuracy spectrophotometer for accurate analysis and transmission of laboratory color.Apply in paints, inks, textiles, garments, printing and dyeing, printing etc industries for color transfer and quality control, also for Fluorescence sample color measurement.	Measuring Aperture	Dual Aperture: MAV: 10mm/8mm, SAV: 5mm/4mm, 1*3mm(some for one aperture)
Integrating Sphere Size	48mm	Specular Component	SCI&SCE
Light Source	Combined LED Light, UV Light(some models have)	Color Space	CIE Lab, XYZ, Yxy, LCh, CIE LUV, Hunter LAB
Spectrophotometric Mode	Concave Grating	Color Difference Formula	ΔE^*ab , ΔE^*uv , ΔE^*94 , $\Delta E^*cmc(2:1)$, $\Delta E^*cmc(1:1)$, ΔE^*00v , $\Delta E(\text{Hunter})$
Sensor	256 Image Element Double Array CMOS Image Sensor	Other Colorimetric Index	WI(ASTM E313, CIE/ISO, AATCC, Hunter), YI(ASTM D1925, ASTM 313),TI(ASTM E313, CIE/ISO), Metamerism Index MI, Staining Fastness, Color Fastness, Color Strength, Opacity, 8 °Glossiness

2. Although this was not stated in my original review, and has no bearing on the scientific merit of the article, I do have an issue with one aspect of the nomenclature used in the manuscript. The authors use the term “E-ink” in this article as a generic term for electrophoretic displays. However, I strongly object to the term “E-ink” as a generic technology descriptor. “E INK” is a trademark by the E Ink Corporation, first filed in 2006. They have several related trademarks, and have successfully prevented other companies using “E Ink” as a generic technology descriptor through legal action. In this case, the authors could use “electrophoretic displays” or “electronic paper” as generic technology terms. I would note that the authors are not solely responsible for using this term, and there are other papers in the recent literature that also use “E-ink” in the same way – however, this is not a valid reason to perpetuate the mistake. Again, as this does not bear on the scientific validity of the work, I will leave it to the editors and the authors to weigh the validity of my argument.

--- The term “E-ink” was used at 82 positions in manuscript R1 and 39 positions in supporting information R1. As requested, they were replaced by “bistable ERPC”, “electrophoretic ink”, or “electrophoretic display (EPD)” contextually. Only 2 “E-ink” terms were kept in the revised manuscript. Please check the revisions in manuscript R2 and supporting information R2.

Reply to Reviewer #4:

1. *I'm afraid I disagree with the author's answer to question #3. Authors claim that if this PC layer has a static pattern composed of fixed RGB PC structures, such a “bilayer” device is not a real reflective display because it can only show an unchanged image. However, that is not true. Simultaneously manipulating color and greyscale expression can provide any images, although the quality of the reflective display is decided by reflectivity (brightness), color gamut, switching speed, pixel resolution, etc.*

This manuscript has shown a new concept of switching them on- and off-state of ERPC with the definite sacrifice of color tunability. Regarding reflectivity (brightness) of the reflective display, color-tunable pixels with ERPC can usually provide a much brighter image, especially in expressing R, G, and B primary colors, because this device does not require shutting down any pixels to express any color. In terms of these facts, the authors should correct the sentence starting with “Similar to” (2nd paragraph on page 3) because readers and other researchers will misunderstand that this technology has both color tunability and switchability, which is not true in this manuscript.

In addition, using any shutters with PC color pixels can provide more switching speed benefits; on the contrary, the manuscript shows poor switching speed due to solvent viscosity, but I think this manuscript still has strong potential due to the first demonstration of the on-off switching concept and a nice device demonstration of ERPC after proper further revision.

--- Inspired by the “simultaneous manipulation of color and greyscale”, we just realized that the coupling of “fixed RGB PC structure” with “optical shutters” having bistability and fast switching ability was a feasible strategy to display any color images. For example, neighboring optical shutters in small size can be grouped to cover a subpixel PC structure to realize a specific color (R/G/B) with different greyscales. Then, the greyscale of RGB subpixels can be tuned independently to output a “colored” pixel via color mixing. Finally, the display panel composed of such pixel arrays can display any color images controlled by electrical signals.

We are sorry for making a false conclusion in the previous reply. As indicated by the reviewer, the bilayer strategy provides a good way to significantly improve the color switching speed. However, such display device based on “fixed RGB PC layer” and “optical shutter” is intrinsically different from the electrophoretic display in this work. The current work focused more on the innovation of PC material but not the flexible using of optical shutter. Due to above considerations, we only mention the bilayer strategy briefly in the discussion section to give readers more inspiration for further researches. Please check the revisions on Page 13-14

As for the misleading in color tunability for the current technology, we had revised the 2nd paragraph on Page 3 according to the reviewer’s suggestion. Now, it clearly states that the bistable ERPC lacks color tunability but possesses switchable bistability. Related revisions are excerpted as below:

In this work, we shall report the first ERPC with bistable and switchable characteristics and develop a new kind of electrophoretic ink for color display. Unlike the conventional ERPC, the SiO₂/PCb-PEG-EG ERPC is unable to show abundant colors at different voltages due to its weak color tunability. However, in addition to the stable “colored state” from colloidal self-assembly, it also has a stable “colorless state” at 0V with disordered particle stacking. Because the introduction of polyethylene glycol (PEG) increases the viscosity to balance the electrostatic repulsion, which further freezes the particles’ Brownian motion and stabilizes the disordered structure.

Reviewer comments, third round-

Reviewer #1 (Remarks to the Author):

I am satisfied with the author's revised version, and have no further comments. I will be happy to see this article in print!

Reviewer #4 (Remarks to the Author):

I think this manuscript is now ready to be published.